# CT-Based Evaluation of Volumetric Posterior Pelvic Bone Density with Implications for the Percutaneous Screw Fixation of the Sacroiliac Joint

**DOI:** 10.3390/jcm13206063

**Published:** 2024-10-11

**Authors:** Michał Kułakowski, Karol Elster, Michał Janiak, Julia Kułakowska, Paweł Żuchowski, Rafał Wojciechowski, Marta Dura, Marcin Lech, Krzysztof Korolczuk, Magdalena Grzonkowska, Michał Szpinda, Mariusz Baumgart

**Affiliations:** 1Clinical Department of Orthopedics and Traumatology, Jan Biziel University Hospital No. 2 in Bydgoszcz, 85-168 Bydgoszcz, Poland; karol.elster@gmail.com (K.E.); michal_dr@interia.pl (M.J.); 2Faculty of Medicine and Health Sciences, Medical College, Andrzej Frycz Modrzewski Krakow University, 30-705 Krakow, Poland; kulakowskajulia@wp.pl; 3Clinic of Rheumatology and Connective Tissue Diseases, Jan Biziel University Hospital No. 2 in Bydgoszcz, Collegium Medicum in Bydgoszcz, 85-168 Bydgoszcz, Poland; pawel.zuchowski@cm.umk.pl (P.Ż.); r.wojciehcowski@wp.eu (R.W.); 4Department of Radiology, Jan Biziel University Hospital No. 2 in Bydgoszcz, Collegium Medicum in Bydgoszcz, 85-168 Bydgoszcz, Poland; martadura83@gmail.com; 5Clinical Department of Orthopedics, Traumatology and Hand Surgery, Jan Mikulicz-Radecki University Clinical Hospital in Wroclaw, 50-556 Wroclaw, Poland; marcinlecj2@wp.pl; 6Department of Orthopedics, Traumatology and Hand Surgery, Faculty of Medicine, Wroclaw Medical University, 50-556 Wroclaw, Poland; krzysztof.korolczuk@usk.wroc.pl; 7Department of Normal Anatomy, The Ludwik Rydygier Collegium Medicum in Bydgoszcz, The Nicolaus Copernicus University in Toruń, 87-100 Toruń, Poland; m.grzonkowska@cm.umk.pl (M.G.); mariusz.baumgart@cm.umk.pl (M.B.); 8Faculty of Medicine, Bydgoszcz University of Science and Technology, 85-796 Bydgoszcz, Poland; michal.szpinda@cm.umk.pl

**Keywords:** iliosacral screw, sacroiliac joint, bone mineral density, pelvis

## Abstract

**Background:** Operative treatment of fragility fractures of the pelvis has become a gold standard. Preoperative planning, including the assessment of the pathway for iliosacral screws, is crucial. The anchorage of the screw depends on the bone quality. Some recent studies have concentrated on assessing bone mineral density (BMD) with the use of Hounsfield unit (HU) values obtained from CT scans. The aim of the present study is to determine the best sacral levels of S1–S3 on the pathway of iliosacral screws for sacroiliac joint fixation. **Methods:** Patients admitted to the Independent Public Healthcare Center in Rypin between 1 of September and 1 of December in 2023, who had CT scans of the pelvis performed on them for different reasons, were included in this study. In total, 103 patients—56 men and 47 women—were enrolled in the study and consecutively separated into two groups of different ages: 18–60 years old (group A) and above 60 years old (group B). The volumetric bone density expressed in HU values was measured with sacral levels of S1, S2 and S3. Apart from the bodies of sacral vertebrae S1–S3, our measurements involved the ala of the ilium in the vicinity of the sacroiliac joint and the wing of the sacrum. All the measurements were performed on the pathway of presumptive iliosacral screws to stabilize the sacroiliac joint. **Results:** In group A (58 patients) the highest bone density in sacral bodies was found in S1 that gradually decreased to S3, while the opposite tendency was demonstrated in the ala of ilium. The HU values in the wing of the sacrum did not display statistical significance. In group B (45 patients), the highest bone density was also found in the sacral body S1 that decreased toward S3 but in the ala of ilium, the highest bone density was found with level S1 and lowest with level S2. In both groups, the highest bone density referred to the wing of the sacrum. **Conclusion:** While the perfect construct for posterior pelvic ring fixation remains unclear, our findings may imply that sacroiliac joint screws inserted into the wing of the sacrum of greater bone density could provide much more successful fixation in comparison to those anchored in the body of sacral vertebra of lesser bone density.

## 1. Introduction

As a very common disease affecting the elderly, osteoporosis results in poor bone density, microarchitecture and strength, thus predisposing this population to fragility fractures [1]. It was estimated that in the year 2000, there were 9 million new fragility fractures, and more than 50 million people worldwide suffered from consecutive sequelae of fractures [2]. Operative treatment for osteoporotic bone fractures remains a gold standard among orthopedic surgeons [3]. Well-established fixation methods for posterior pelvic ring injuries remain in the use of both iliosacral and transsacral screws. A thorough understanding of injuries—involving available osseous fixation pathways, the appropriate preoperative planning, apposite interpretation of intraoperative imaging and precise surgical techniques—requires the ability to safely perform iliosacral screw fixation for the sacroiliac joint [4,5].

In orthopedic surgery, a key risk factor of fragility fractures results from impaired bone quality. It is noteworthy that both younger and older patients are prone to pelvic fractures [6]. In younger patients, a severe high-energy trauma incident is required to provoke fractures of the pelvic ring, whereas in the elderly with poor bone quality, even a low-energy trauma incident may lead to pelvic fractures [7]. Indispensably, in order to achieve the best possible operative outcome, orthopedic surgeons have to both understand the mechanism of a particular trauma and fine-tune the preoperative planning.

Some recent studies have concentrated on bone mineral density (BMD) with the use of Hounsfield unit (HU) values obtained from CT scans [8,9]. Furthermore, HU values have also been used to evaluate BMD in the diagnosis of osteoporosis. These studies are based on CT scans of the distal radius, the head and neck of the femur and the proximal humerus [10,11].

To the best of our knowledge, there exists only one article that examined the CT- based bone density of the ilium and the sacrum regardingiliosacral screw fixation for the sacroiliac joint. According to the professional literature, sacral vertebra 1 is characterized by the greatest CT-based bone density. Due to dysmorphic sacral vertebra 1, some studies have focused on the usefulness of the more distal sacral segments for sacroiliac joint fixation. Although the availability of the sacral levels caudal to the first sacral vertebra for sacroiliac joint fixation has already been established, the evidence value of screws in this area is still questioned in the professional literature due to the standpoint of minor bone quality [4]. On the other hand, sacral vertebra 2 has been reported as a usual pathway for inserting iliosacral screws [6].

The aim of the present study is to determine the best sacral levels of S1–S3 on the pathway of iliosacral screws for sacroiliac joint fixation, at which the ala of the ilium and the wing of the sacrum are of the relatively greatest bone density. Our main purpose is to examine the differences in bone density expressed in HU values to receive both the best possible outcome and to enable the most effective preoperative planning to reduce loose screws.

Contrary to recent studies, we hypothesize that screws inserted into sacral vertebra 2 result in the best possible operative outcome.

## 2. Materials and Methods

### 2.1. Population

The present retrospective study was conducted in accordance with the Declaration of Helsinki and approved by the Bioethical Committee of the Kuyavian-Pomeranian Local Medical Chamber.

Patients were admitted to our institution, i.e., Independent Public Healthcare Center in Rypin, in 2023 between September 1 and December 1. The inclusion criteria were skeletal maturity, the patient’s age being over 18 years and recommendation for the patient to obtain abdomen or pelvis CT scans. Patients who had suffered from pelvic traumas, malignancy, rheumatoid arthritis or steroid medication were excluded from the study. Finally, 103 patients—56 men and 47 women—were enrolled in the study. The patients’ ages ranged from 18 to 90 years, with a mean age of 55.9 years. To investigate the potential impact of increased sclerosis with age and, theoretically, a decrease in bone density, the patients were separated into two groups: group A encompassed patients 18–60 years old, while group B involved patients over 60 years. Group A comprised 58 patients: 31 men and 27 women, with a mean age of 42.7 years. Group B consisted of 45 patients: 25 men and 20 women, with a mean age of 72.8 years).

### 2.2. Technique Procedures

Abdomen or pelvis CT scans were performed using Siemens Somatomgo.Up (Siemens, Munich, Germany). Patients’ demographic data, including age and gender, from the system were collected.

The abdomen or pelvis CT scans of each patient were assessed in an identical fashion by one investigator (JK) specializing in pelvic traumas with the use of the picture archiving communication system (PACS).

Images were viewed using the bone algorithm default windows. The investigator first determined axial images in levels of mid-lengths of bodies of sacral vertebrae 1, 2 and 3 that afterward were confirmed by coronal and sagittal reconstructions (Figure 1, Figure 2 and Figure 3). The volumetric bone density expressed in HU values was measured in sacral levels of S1, S2 and S3. Apart from the bodies of sacral vertebrae S1–S3, the measurements involved the ala of the ilium in the vicinity of the sacroiliac joint and the wing of the sacrum. All the measurements were performed on the pathway of presumptive iliosacral screws to stabilize the sacroiliac joint.

In order to standardize measurements while accounting for normal anatomic variations and optimum iliosacral screw trajectories—so as to avoid injuries of neurovascular structures—standardized circular voxel regions of interest (ROIs) were drawn (Figure 1, Figure 2 and Figure 3) [12]. As presented in the former studies, these ROIs were drawn in levels of mid-lengths of the bodies of sacral vertebrae 1, 2 and 3, the right and left wings of sacrum and the ala of the ilium in sacral levels of S1–S3 [13]. The aforementioned standardized ROIs were drawn with areas ranging from 0.8 to 1.2 cm^2^. This range of area was estimated as it best represented the osseous surface area, which is required to safely place 7.0–mm iliosacral or transsacral screws [4].

In sacral levels S1, S2 and S3, the bone density was measured at the following five points: mid-length of the body of the sacral vertebra, right and left wings of the sacrum in the vicinity of the sacroiliac joint and the ala of the ilium on the right and left sides in the vicinity of the sacroiliac joint. There were five measurements at each sacral level and fifteen measurements in each patient.

### 2.3. Statistical Analysis

The statistical analysis was performed using the PQ Stat 1.8.6 (Poznan, Poland). Because all numerical data displayed a normal distribution, our results have been presented as means with the standard deviations.

## 3. Results

No significant right–left differences were found in the bone density expressed in HU values for the ala of the ilium and the wings of the sacrum (*p*> 0.05).The mean values with standard deviations of the bone density expressed in HU values for the bodies of sacral vertebrae S1–S3 and associated sacral levels of the wing of sacrum and the ala of ilium have been presented in Table 1 for group A and Table 2 for group B.

### 3.1. Bone Density in Group A

In group A, with sacral level S1, the mean values of the bone density expressed in Hounsfield units for the body of sacral vertebra S1, the wing of the sacrum and the ala of ilium reached the following values: 194.9 HU, 388.9 HU and 81.45 HU, respectively. Correspondingly, these values were 134.4 HU, 376.6 HU and 145.85 HU with sacral level S2 and 99.6 HU, 365.7 HU and 220.35 HU with sacral level S3.

The bone density for the bodies of sacral vertebrae S1–S3 expressed in Hounsfield units followed a decreasing sequence: 194.9 ± 86.4 HU, 134.4 ± 94.3 HU and 99.6 ± 74.18 HU. The difference in bone density was statistically insignificant between the bodies of sacral vertebrae S1 and S2 (*p* = 1.4). Contrariwise, these differences were statistically significant between the bodies of sacral vertebrae S1 and S3 (*p* < 0.001) and between the bodies of sacral vertebrae S2 and S3 (*p* = 0.03).

The bone density for the wing of the sacrum with sacral levels S1, S2 and S3 reached the values 388.9 HU, 376.6 HU and 365.7 HU, respectively. The differences in bone density were statistically insignificant between sacral levels S1 and S2 (*p* = 9.7) and between sacral levels S2 and S3 (*p* = 0.15). This difference was statistically significant between sacral levels S1 and S3 (*p* = 0.04).

The bone density for the ala of ilium with sacral levels S1, S2 and S3 were 81.45 HU, 145.85 HU and 220.35 HU, respectively. There were statistically insignificant differences between sacral levels S1 and S2 (*p* = 4.9) and between sacral levels S2 and S3 (*p* = 3.3). However, this difference proved to be statistically significant between sacral levels S1 and S3 (*p* < 0.001).

### 3.2. Bone Density in Group B

As far as group B is concerned, with sacral level S1, the mean values of the bone density for the body of sacral vertebra S1 was 212.87 HU, for the wing of the sacrum was 395.90 HU and for the ala of the ilium was 233.55 HU. The respective numerical data were 107.12 HU, 339.00 HU and 124.96 HU with sacral level S2 and 85.32 HU, 360.05 HU and 199.91 HU with sacral level S3. As presented, in the caudal direction, the bone density for the bodies of sacral vertebrae S1–S3 expressed in Hounsfield units revealed a successive decrease in values. Of note, the differences in bone density were statistically significant between the bodies of sacral vertebrae S1 and S2 (*p* < 0.001) and between the bodies of sacral vertebrae S1 and S3 (*p* < 0.001). However, this difference between the bodies of sacral vertebrae S2 and S3 (*p* < 0.001) was not statistically significant (*p* = 0.08). The bone density for the wing of the sacrum with sacral levels S1, S2 and S3 reached the values 395.90 HU, 339.00 HU and 360.05 HU, correspondingly. The differences in their bone density were of statistical significance between sacral levels S1 and S2 (*p* < 0.001) and between sacral levels S1 and S3 (*p* = 0.014). This difference was statistically insignificant between sacral levels S2 and S3 (*p* = 0.12).

With relation to the ala of ilium, its bone density was 233.55 HU for sacral level S1, 124.96 HU for sacral level S2 and 199.91 HU for sacral level S3. There were statistically significant differences between sacral levels S1 and S2 (*p* < 0.001) and between sacral levels S2 and S3 (*p* < 0.001). There was no statistical difference between sacral levels S1 and S3 (*p* = 0.21).

In comparison to group A, the different tendency for the bone density of the ala of the ilium was found in group B.

## 4. Discussion

Percutaneous iliosacral screw fixation is regularly used as a treatment of choice for unstable pelvic ring injuries [14]. Since the posterior pelvic anatomy remains complex and variable, percutaneous pelvic ring fixation may technically be challenging. Careful preoperative planning, including both the pelvic anatomy and bone density, is indispensable for determining the best corridor for safe iliosacral screw fixation [15]. In the medical literature, the incidence of dysmorphism of the sacrum has been reported to be up to 44%, so a thorough understanding of the anatomy of the sacrum is critical for safely placing sacroiliac joint screws [16,17]. Furthermore, the safe corridor for iliosacral screw fixation with sacral level S2 was found to be in 95% of dysmorphic sacra and only in 50% of normal ones [16,18].

The optimum fixation construct of the posterior pelvic ring using sacroiliac joint screws remains unclear, although some biomechanical studies proposed the improvement of stability using two points of fixation [19]. Chatain et al. recommended using implants of larger diameter for fixation of the posterior pelvic ring [20]. While searching for the best method of stabilizing the posterior pelvic ring, some previous studies reported benefits and increased strength of transsacral screws in comparison with iliosacral screws due to multiple points of cortical purchase [21]. Due to the safe corridor, some authors emphasized the ability and safety of transsacral screws with sacral level S2, at which surgeons commonly stabilize posterior pelvis ring injuries [22,23]. Salazar et al. reported the bone density of the body of sacral vertebra S1 to be significantly greater than that of the body of sacral vertebra S2 [13]. Consequently, these authors advised surgeons not to use the safe corridor into the body of sacral vertebra S2 due to its decreased bone density. Regrettably, the bone density of the ala of the ilium and the wing of the sacrum was not reported in their study. Another study by Eastman et al. found the greatest bone density of the posterior pelvic ring, including both the sacroiliac joint and the ala of ilium, to be with sacral level S2 [4].

Having taken into account disparate findings offered by the aforementioned authors, in the present study, we hypothesize that from a bone quality perspective, the body of sacral vertebra S2 is useful for the safe corridor while inserting sacroiliac joint screws.

In accordance with the former studies to examine the bone density with the use of HU values, we do so with relation to the bodies of sacral vertebrae S1–S3, the wing of the sacrum and the ala of ilium at the corresponding sacral levels on the pathway of sacroiliac joint screws [6,24,25].

In group A with patients aged 18–60 years old, we found the bone density of the bodies of sacral vertebrae S1 and S2, the wing of the sacrum and the ala of ilium with sacral levels S1 and S2 to be similar with statistically insignificant differences. Thus, our findings are somewhat different from those presented by Salazar et al. and Eastman et al. [4,13]. The authors found the difference in bone quality between S1 and S3 to be statistically significant. Consequently, we do not recommend inserting sacroiliac joint screws at sacral level S3. Our riveting finding is that the wing of the sacrum in group A reveals the greatest bone density compared with all three sacral levels S1–S3. This is incongruent with the results offered by Schonenberg et al., who evaluated the bone quality in patients over 50 years with fragility fractures [6].

In the present study, we found that in both groups A and B with sacral levels S1–S3 that the wing of the sacrum was of a greater bone density than the corresponding bodies of sacral vertebrae. This is the reason why we recommend the anchorage of sacroiliac joint screws into the wing of the sacrum rather than into the bodies of sacral vertebrae. From this aspect, we agree with the findings of some former studies [26,27].

In our group B comprising patients over 60 years old, we proved the tendency of decreasing bone quality of the sacrum from sacral level S1 towards sacral level S3. This referred to both the bodies of the sacrum and the wing of the sacrum and was similar to the results from group A. As with the findings presented by Eastman et al. and Schonenberg et al., in the present study, the difference in bone density between sacral levels S1 and S2 was statistically significant [4,6].

Some studies have exposed that sacral level S3 is another osseous fixation pathway of potential relevance [28,29]. Since our results have demonstrated that the wing of the sacrum displaysthe greatest bone density at three sacral levels, S1–S3, compared with the sacral body and sacral ala, the wing of the sacrum with sacral level S3 may successfully be useful for the insertion of sacroiliac joint screws. It is noteworthy to emphasize that the body of sacral vertebra 3 presents a statistically significant decrease in bone density.

A considerable limitation of our study is that HU values were used as a substitute for the evaluation of bone density. Even though HU values have successfully been used in previous studies, HU values still provide only the quantitative evaluation of bones, not indicative of the underlying bone quality, and CT scans were not calibrated for BMD [7,13,30]. Another limitation of this study may be that only one author performed all measurements of bone density.

## 5. Conclusions

In conclusion, while the perfect construct for posterior pelvic ring fixation remains unclear, our findings may imply that sacroiliac joint screws inserted into the wing of the sacrum of greater bone density could provide much more successful fixation in comparison to those anchored in the body of sacral vertebra of lesser bone density. Our study demonstrates comparable bone density in the bodies of sacral vertebrae 1 and 2 and relatively greater bone density in the wing of the sacrum with sacral levels S1–S3. The main difference in both groups refers to the bone density of the ala of ilium, which is greater in the elderly. This may result from degenerative changes around the sacroiliac joint, mostly caused by the osteoarthritis detected in some patients in group B.

## Figures and Tables

**Figure 1 jcm-13-06063-f001:**
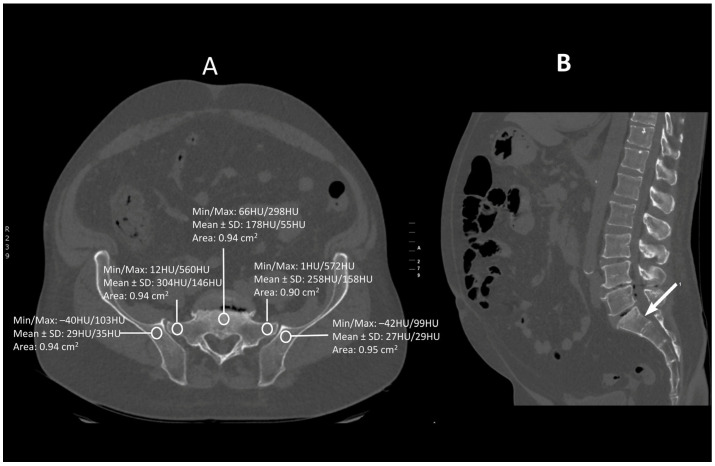
A transverse (**A**) and saggital (**B**) projection of CT scan in level S1. A circular region of interest (ROI) ranging from 0.8 to 1.2 cm^2^ has been placed at the ala of the ilium, the wing of the sacrum and the body of sacral vertebra 1.

**Figure 2 jcm-13-06063-f002:**
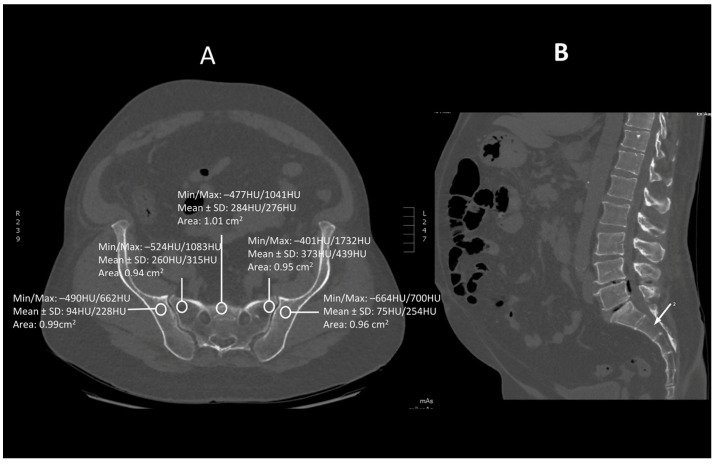
A transverse (**A**) and saggital (**B**) projection of CT scan in level S2. A circular region of interest (ROI) ranging from 0.8 to 1.2 cm^2^ has been placed at the ala of the ilium, the wing of the sacrum and the tbody of sacral vertebra 2.

**Figure 3 jcm-13-06063-f003:**
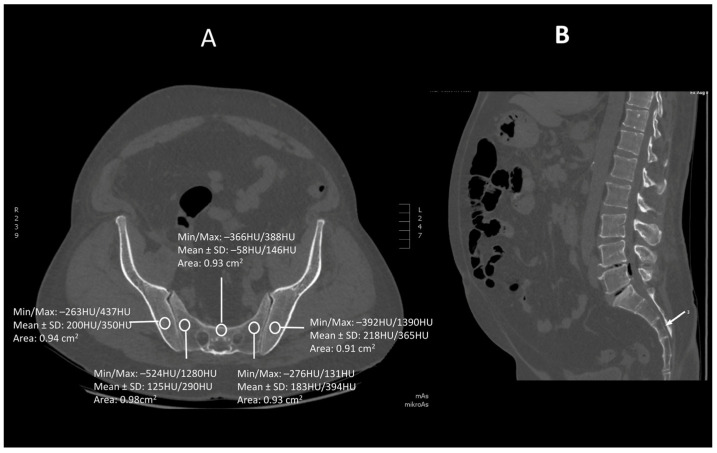
A transverse (**A**) and saggital (**B**) projection of CT scan in level S3. A circular region of interest (ROI) ranging from 0.8 to 1.2 cm^2^ has been placed at the ala of the ilium, the wing of the sacrum and the body of sacral vertebra 3.

**Table 1 jcm-13-06063-t001:** The mean HU values with standard deviations of the bone density for the bodies of sacral vertebrae S1–S3 and associated sacral levels of the wing of the sacrum and the ala of ilium for patients 18–60 years old.

	S1	S2	S3	*p*
Body of sacral vertebra	194.9 ± 86.4 HU	134.4 ± 94.3 HU	99.6 ± 74.18 HU	S1 vs. S2 = 1.4 S2 vs. S3 = 0.03 S1 vs. S3 < 0.001
Wing of sacrum	388.9 ± 110.05 HU	376.6 ± 100.29 HU	365.7 ± 120.5 HU	S1 vs. S2 = 9.7 S2 vs. S3 = 0.15 S1 vs. S3 = 0.04
Ala of ilium	81.45 ± 64.32 HU	145.85 ± 97.45 HU	220.35 ± 93.0 HU	S1 vs. S2 = 4.9 S2 vs. S3 = 3.3 S1 vs. S3 < 0.001

**Table 2 jcm-13-06063-t002:** The mean HU values with standard deviations of the bone density for the bodies of sacral vertebrae S1–S3 and associated sacral levels of the wing of the sacrum and the ala of ilium for patients over 60 years old.

	S1	S2	S3	
Body of sacral vertebra	212.87 ± 78.75 HU	107.12 ± 75.60 HU	85.32 ± 68.26 HU	S1 vs. S2 < 0.001 S2 vs. S3 = 0.08 S1 vs. S3 < 0.001
Wing of sacrum	395.90 ± 138.72 HU	339.00 ± 110.66 HU	360.05 ± 108.84 HU	S1 vs. S2 < 0.001 S2 vs. S3 = 0.12 S1 vs. S3 = 0.014
Ala of ilium	233.55 ± 177.91 HU	124.96 ± 69.21 HU	199.91 ± 99.46 HU	S1 vs. S2 < 0.001 S2 vs. S3 < 0.001 S1 vs. S3 = 0.21

## Data Availability

The original contributions presented in the study are included in the article, further inquiries can be directed to the corresponding author.

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
