# Peer review of "CT-Based Evaluation of Volumetric Posterior Pelvic Bone Density with Implications for the Percutaneous Screw Fixation of the Sacroiliac Joint"

_jcm, 2024, doi:10.3390/jcm13206063_

Round 1

Reviewer 1 Report

Comments and Suggestions for Authors

This article focuses on the measurement bone mineral density over pelvis. The data can provide useful trajectory of sacrum-lilac screw insertion. There are few questions here.

1.        About the HU value between group 1 and 2, HU vales in group 2(> 60 years old) are the same or even greater than group1(< 60 years old), which is very unusual. Can authors provide supplement data or further explanation.

2.        Authors have demonstrated the ROI (region of interest) over ala of ilium representing the trajectory of sacroiliac screw. But it not the usual entry point of SI screw. According to previous research (citation 4 and 13), the normal ROI are more anteriorly when compared to current study. Authors should explain why you chose this point.

3.        The most complicated cases during inserting SI screw are patients with S1 dysmorphism. Is there any data about the bony density of S1 dysmorphism in this study. Bony density in S1 dysmorphism is higher or lower than normal population and can authors provide data to verify the more precise trajectory of SI screw in sarcum dysmorphism.

Comments on the Quality of English Language

 There are a lot of grammatical error through the whole articles, they should be revis

Reviewer 2 Report

Comments and Suggestions for Authors

-              How did you choose to split the age groups? Why 18-60 years old and 60 years and above?

-              Reword lines 56-59. Suggest “A thorough understanding of injuries pre-requisites the ability to safely perform iliosacral screw fixation for the sacroiliac joint. This involves knowledge of available osseous fixation pathways…”

-              Line 82: Do you mean aim instead of assumption? 

-              Line 101: do you mean decrease in bone density? Theoretically there is a decrease in bone density with age, not an increase.

-              Throughout the text please replace “with levels of mid-length of the bodies of sacral vertebrae” with “in level with …”. 

-              Please give your Mean and standard deviation values in figures 1-3 in the form of Mean ± SD

-              Line 248: Do you mean ‘compared with all three sacral levels S1-S3’ instead of ‘with’. 

-              Line 256: Group B has patients ages over 60 years, but you have written 65 years old here. Is this a typo?

-              In the limitations section it is worth noting that the CT scans were not calibrated for BMD. 

-              Line 279: are you sure that the bone density of the Ala of the ilium is greater in the elderly? This is only true at the level of S1, when looking at the values presented in table 1 and 2.

-              How many patients have osteoarthritis (OA) in both age groups? Please provide this information as this can also be a factor that could affect the results and must be considered? Perhaps separating the patients into those with and without OA is another part of this study that could lead to relevant clinical findings.

Comments on the Quality of English Language

The quality of the English language in this paper can be improved as there are some sentences which can be broken up into two. Additionally, some phrases need rewording for greater clarity. But, overall it is not a difficult read.
